# JustLogic: A benchmark for natural language deductive reasoning

## Abstract

Logical reasoning is a critical component of Large Language Models (LLMs), and substantial research efforts in recent years have aimed to enhance their deductive capabilities. However, existing deductive reasoning benchmarks, which are crucial for evaluating and advancing LLMs, are inadequate due to their lack of task complexity, presence of prior knowledge as a confounder, and superficial error analysis. To address these deficiencies, we introduce JustLogic, a synthetically generated deductive reasoning benchmark designed for rigorous evaluation of LLMs. JustLogic is (i) highly complex, capable of generating a diverse range of linguistic patterns, vocabulary, and argument structures; (ii) prior knowledge independent, eliminating the advantage of models possessing prior knowledge and ensuring that only deductive reasoning is used to answer questions; and (iii) capable of in-depth error analysis on the heterogeneous effects of reasoning depth and argument form on model accuracy. Our experimental results on JustLogic reveal that the performance of most state-of-the-art (SOTA) LLMs, specifically Llama3-8B (57.8%), Llama3-70B (64.6%), and GPT-4o (65.6%), is significantly worse than the average human performance (73.0%). A recently released reasoning model, OpenAI o1-preview, performed substantially better, with an accuracy of 81.0%. However, it still lags behind the human ceiling of 100.0%. These results demonstrate that the JustLogic benchmark is realistic and achievable for both humans and models and that there is still substantial room for improvement in the deductive reasoning capabilities of LLMs. We posit that the use of prior knowledge dependent and relatively simplistic benchmarks has misrepresented the reasoning abilities of many SOTA models. We release our open-source dataset to provide accurate evaluations of model performance in deductive reasoning and to facilitate LLM advancement through in-depth error analysis.[1]

## 1 Introduction

Deductive reasoning is a crucial capability for large language models (LLMs). It refers to the process of creating logically valid arguments, where conclusions necessarily follow from the premises. In other words, if an argument's premises are true, the conclusion must also be true. Recent state-of-the-art (SOTA) LLMs (Achiam et al., 2023; Dubey et al., 2024; Jiang et al., 2023) have exhibited outstanding performance and consistent improvement across various reasoning benchmarks, including HelloSwag (Zellers et al., 2019), ARC Challenge (Clark et al., 2018) and WinoGrande (Sakaguchi et al., 2021). However, we argue that the existing benchmarks are insufficient and often ineffective for evaluating LLMs' true deductive reasoning capabilities.

We identify three major problems with the existing benchmarks. **First**, they lack complexity, which is measured on two dimensions: natural language complexity, which refers to how arguments are linguistically expressed, and argument complexity, which pertains to the structure of the argument itself. Manually curated datasets, such as FOLIO (Han et al., 2022) and LogiQA 2.0 (Liu et al., 2020; 2023a) exhibit high natural language complexity but low argument complexity, while synthetic datasets like CLUTRR (Sinha et al., 2019) and ProofWriter (Tafjord et al., 2020) show the opposite. Simplicity in either dimension makes these benchmarks prone to overfitting and memorization, thus allowing models to perform well despite underlying weaknesses in logical reasoning.

---

[1]All code and data are available at https://anonymous.4open.science/r/JustLogic

A more detailed analysis can be found in Section 3.4. **Second**, existing benchmarks often fail to test deductive reasoning in isolation, as models can benefit from prior knowledge. To empirically validate this claim, we developed a novel test for prior knowledge independence, which measures the influence of prior knowledge on reasoning benchmarks. As detailed in Section 5.1, prior knowledge can substantially increase accuracy, even in datasets not intended to require commonsense or domain knowledge, e.g. FOLIO and LogiQA 2.0. Thus, high accuracy may not reflect strong reasoning capabilities. **Third**, many existing benchmarks provide superficial error analysis, leaving key questions unanswered: At what reasoning depth does the model start to fail? How does the model compare to humans at different argument depths? Which argument forms is the model particularly weak at? These insights are essential for understanding the depth and robustness of a model's deductive reasoning, yet many benchmarks fail to provide such insights due to their construction methods. Section 5.3 demonstrates the importance and usefulness of comprehensive error analysis.

Due to these issues, it remains unclear whether deductive reasoning abilities have genuinely advanced despite improving performance on various benchmarks. In response to the critical need for a reliable benchmark to support ongoing research efforts, we present JustLogic, a novel natural language deductive reasoning benchmark. Each instance in JustLogic contains a paragraph of premises and a statement. The task is to determine whether the statement is true, false, or uncertain, based solely on the premises, and assuming they are all true. An example is shown in Figure 1.

---

**Paragraph:**
- Whenever it is true that night blooming plants and trees depend on nectar eating bats for pollination, 'if many species are critically endangered, then it is not true that doors are solids' is true.
- Night blooming plants and trees depend on nectar eating bats for pollination.
- We can assume that many species are critically endangered.

**Question:** Is the following statement true, false, or uncertain?
**Statement:** Doors are solids.
**Answer:** False

---

Figure 1: Example of a question adapted from the JustLogic train dataset

JustLogic's construction ensures it is (i) complex, (ii) prior knowledge independent, and (iii) capable of in-depth error analysis. **First**, to achieve high complexity in both argument structures and natural language, JustLogic is code-generated rather than manually curated. This allows the generation of a theoretically infinite number of unique argument structures. Natural language sentences are then drawn from GenericsKB-Best (Bhakthavatsalam et al., 2020), a database of 1M+ unique sentences, and inserted into the argument structures, introducing high natural language complexity. **Second**, since sentences are randomly sampled from the entire GenericsKB-Best dataset, the generated arguments generally do not align with real-world knowledge, thereby eliminating the influence of prior knowledge and ensuring prior knowledge independence. **Finally**, in-depth error analysis is enabled by our programmatic generation process, which allows us to inspect detailed properties of each question, such as reasoning depth and argument form, and investigate their impact on model performance. A comparison between JustLogic and four similar logical reasoning benchmarks (CLUTRR, ProofWriter, LogiQA 2.0, and FOLIO) is presented in Table 1, with further details on dataset construction provided in Section 3.

Using JustLogic, we conducted comprehensive experiments to evaluate the deductive reasoning capabilities of current LLMs. First, our novel prior knowledge independence test demonstrated that prior knowledge enables LLMs to bypass deductive reasoning on existing datasets, resulting in artificially high accuracies. In contrast, using prior knowledge with JustLogic *reduces* performance, ensuring that results accurately reflect deductive reasoning ability. Second, we benchmarked the performance of SOTA LLMs and human participants using JustLogic. Most SOTA LLMs, regardless of parameter size or prompting method, performed significantly lower than the average human accuracy (73.0%). OpenAI o1-preview performed substantially better (81.0%), but still fell short of the human ceiling (100.0%). Finally, enabled by JustLogic's code-generated nature, our thor-

Table 1: Comparison of JustLogic with other deductive reasoning datasets. The symbol $\sim$ suggests the feature is present but to a limited extent.

|  | High NL Complexity | High Arg. Complexity | Prior Knowledge Independence | In-Depth Error Analysis |
|---|---|---|---|---|
| CLUTRR | ✗ | ✓ | ✓ | ✓ |
| ProofWriter | ✗ | ✓ | ✓ | $\sim$ |
| LogiQA 2.0 | ✓ | ✗ | ✗ | $\sim$ |
| FOLIO | ✓ | ✗ | ✗ | $\sim$ |
| **JustLogic** | ✓ | ✓ | ✓ | ✓ |

ough error analysis examined the impact of various question properties, such as argument structure and reasoning depth, on model performance. These experimental results show that the JustLogic benchmark is both realistic and achievable for humans and models, and reveals significant room for improvement in LLM deductive reasoning capabilities.

In summary, our contributions are threefold. First, we evaluate the limitations of existing benchmarks. Second, we introduce the JustLogic benchmark, a synthetic dataset to evaluate deductive reasoning, that addresses the aforementioned limitations. Third, our experiments using JustLogic demonstrate that most SOTA models perform significantly worse than humans. We posit that the deductive reasoning capabilities of LLMs still have significant room for improvement, and hope that the JustLogic benchmark will assist researchers in designing and evaluating LLMs.

## 2 RELATED WORK

### 2.1 EXISTING REASONING DATASETS FOR LARGE LANGUAGE MODELS

Reasoning benchmarks are a vital part of LLM evaluation. Some benchmarks measure deductive reasoning in conjunction with natural language inference (NLI), inductive reasoning, and commonsense knowledge: HellaSwag (Zellers et al., 2019) tasks machines to select the most likely follow-up of an event description, WinoGrande (Sakaguchi et al., 2021) is a pronoun resolution task, and MuSR (Sprague et al., 2023) tasks machines to solve fictional problems, such as murder mysteries. Other benchmarks measure reasoning on domain knowledge: AI2 Reasoning Challenge (ARC) (Yadav et al., 2019) contains grade-school science questions, while Massive Multitask Language Understanding (MMLU) (Hendrycks et al., 2020) contains questions across 57 subjects in STEM, humanities, and more. Finally, math-specific benchmarks include GSM-8K (Cobbe et al., 2021) and DROP (Dua et al., 2019).

The aforementioned benchmarks explicitly evaluate skills beyond reasoning and do not specifically define the type of reasoning involved, e.g. inductive, deductive, and analogical. As such, benchmarks that solely test for deductive reasoning have seen a considerable increase in interest. They can be classified into two broad categories: synthetic and manually curated. Synthetic datasets include (i) CLUTRR (Sinha et al., 2019), where a machine must infer the relationship of two family members based on stories, (ii) ProofWriter (Tafjord et al., 2020), where a machine must deduce a statement's truth value based on a set of facts and rules, and (iii) ProntoQA-OOD (Saparov et al., 2024), where a machine must prove a statement based on a set of facts. Manually curated datasets include (i) LogiQA 2.0 (Liu et al., 2023a), containing manually-translated logical reasoning questions from the Chinese Civil Service Exam, (ii) FOLIO (Han et al., 2022), containing questions with manually-annotated content using Wikipedia pages, and (iii) ReClor (Yu et al., 2020), containing reading comprehension questions from GMAT and LSAT.

As discussed earlier, synthetic datasets are prior knowledge independent and exhibit high argument and low natural language complexity; manually curated datasets are the opposite. JustLogic, being synthetic, contains all its advantages while offering the natural language complexity of manually curated datasets. Further discussion on JustLogic's complexity and prior knowledge independence can be found in Section 3.4 and 5.1 respectively.

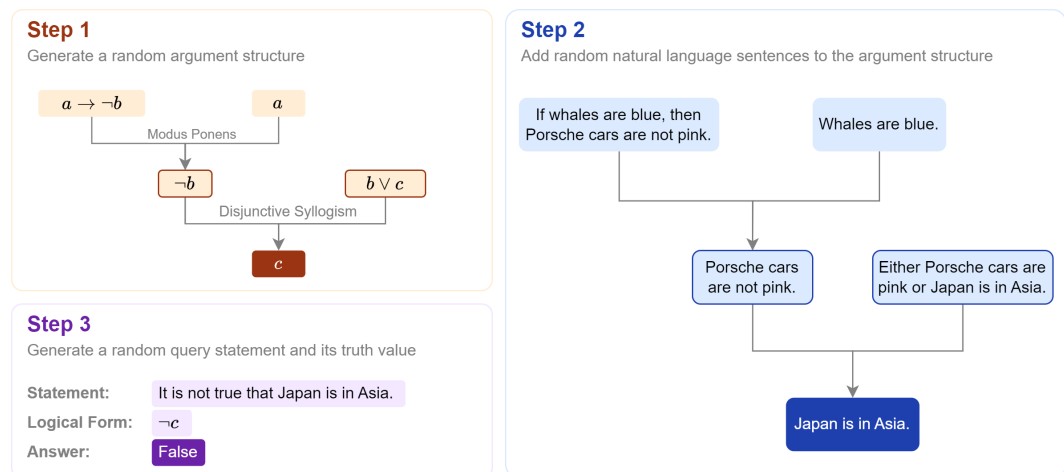

Figure 2: A step-by-step example of how an instance with a reasoning depth of 2 is constructed

## 2.2 REASONING IN LARGE LANGUAGE MODELS

As LLMs continue to increase in size, their performance on various reasoning-related benchmarks has improved dramatically. For example, in 2024, Gemini Ultra (Team et al., 2023) achieved 90.0% on MMLU when the SOTA model in 2020, UnifiedQA 11B (Khashabi et al., 2020), achieved a mere 48.9%. In 2023, GPT-4 achieved 96.4% on ARC when the SOTA model in 2020, GPT-3 (Brown, 2020), achieved 53.2%.

The advent of prompting techniques played an important role in developing LLMs' reasoning abilities. In-context learning (Dong et al., 2022) provides LLMs with instructions and examples in the input prompt to guide its response. Chain-of-thought prompting (Wei et al., 2022) prompts LLMs to generate a series of intermediate reasoning steps before arriving at the final answer. Self-consistency decoding (Wang et al., 2022) chooses the most consistent answer after sampling multiple chain-of-thought outputs. Least-to-most prompting (Zhou et al., 2022) decomposes a complex problem into simpler subproblems, which are then solved sequentially.

As mentioned above, LLMs are conventionally tested on datasets that combine reasoning with other skills. Moreover, existing logical reasoning-specific datasets possess major limitations that call into question the reliability of their evaluations. In response, JustLogic aims to robustly and accurately evaluate the deductive reasoning abilities of LLMs.

## 3 DATASET CONSTRUCTION

JustLogic is a programmatically generated dataset designed to evaluate a model's ability of deductive reasoning, specifically its capability to form logically valid arguments. A logically valid argument is one where the conclusion necessarily follows from the premise(s); in other words, given the premises are true, the conclusion must also be true.

In order to test this, JustLogic presents a model with a paragraph consisting of premises, followed by a query statement. Based solely on the premises and assuming they are all true, the model needs to determine whether the query statement is true, false, or uncertain. In line with the open-world assumption, the "Uncertain" answer refers to cases where the premises neither support nor contradict the query statement.

The following outlines the process for generating each instance in the dataset:

1. Step 1: Generate an argument structure
2. Step 2: Add natural language statements to the argument structure
3. Step 3: Generate a query statement

Figure 2 provides an example of this process, which we reference throughout the rest of this section.

## 3.1 STEP 1: GENERATE ARGUMENT STRUCTURE

Argument structures are composed of one or more valid argument forms, derived from propositional logic; argument forms are made up of a series of logical forms, which we define as symbolic representations of statements. Specifically, the seven distinct argument forms in our dataset are constructed with the following four logical forms: (i) basic ($x$), (ii) negation ($\neg x$), (iii) conditional ($x \rightarrow y$), and (iv) disjunction ($x \vee y$). While there is a theoretically infinite number of possible argument forms, complex argument forms can be derived by combining simpler ones. Therefore, we explicitly define the most fundamental forms (Johnson, 2006), as shown in Table 2.

Table 2: An overview of the argument forms in the JustLogic dataset

|  | Formal Notation | Example |
| --- | --- | --- |
| Modus Ponens | $p \rightarrow q$ | If the sky is blue, then the dog is happy. |
|  | $p$ | The sky is blue. |
|  | $\vdash q$ | Therefore, the dog is happy. |
| Modus Tollens | $p \rightarrow q$ | If the sky is blue, then the dog is happy. |
|  | $\neg q$ | The dog is not happy. |
|  | $\vdash \neg p$ | Therefore, the sky is not blue. |
| Hypothetical Syllogism | $p \rightarrow q$ | If the sky is blue, then the dog is happy. |
|  | $q \rightarrow r$ | If the dog is happy, the owner is happy. |
|  | $\vdash p \rightarrow r$ | Therefore, the owner is happy. |
| Disjunctive Syllogism | $p \vee q$ | Either the dog is barking or the dog is asleep. |
|  | $\neg p$ | The dog is not barking. |
|  | $\vdash q$ | Therefore, the dog is asleep. |
| Reductio ad absurdum | $p \rightarrow q$ | If the dog is calm, the owner is around. |
|  | $p \rightarrow \neg q$ | If the dog is calm, the owner is not around. |
|  | $\vdash \neg p$ | Therefore, the dog is not calm. |
| Constructive Dilemma | $p \vee q$ | Either the sky is blue or it is raining. |
|  | $p \rightarrow r$ | If the sky is blue, the race can start. |
|  | $q \rightarrow s$ | If it is raining, the race is delayed. |
|  | $\vdash r \vee s$ | Therefore, either the race can start or it is delayed. |
| Disjunction Elimination | $p \vee q$ | Either the sky is blue or it is raining. |
|  | $p \rightarrow r$ | If the sky is blue, the dog is cheerful. |
|  | $q \rightarrow r$ | If it is raining, the dog is cheerful. |
|  | $\vdash r$ | Therefore, the dog is cheerful. |

The function to create an argument structure accepts an intended argument depth as input. It first generates a random conclusion and an argument form to support it, which is $c$ and disjunctive syllogism in Figure 2 respectively. If the intended depth has not been reached, one or more premises will become subconclusions, which are supported by new, randomly generated argument forms, thus increasing the argument's depth. In Figure 2, this is exemplified by $\neg b$ being converted to a subconclusion that is supported by a modus ponens argument. If further depth is still required, one or more premises from the newly generated argument forms will themselves have argument forms to support them. This process continues until the target depth is achieved.

## 3.2 STEP 2: ADDING NATURAL LANGUAGE

Once the argument structure is generated, it serves as the skeleton of the paragraph, and the next step is to convert the statements in logical form into natural language. Each statement consists of one or more logical forms, *i.e.* variable, negation, conditional, and disjunction. In natural language, these forms can be expressed in a variety of ways. For example, a conditional can be expressed as both "If $x$, then $y$." and "Given that $x$, $y$ is true.", where variables $x$ and $y$ are simple propositions. To emulate the diversity of natural language, we manually create a list of expressions for each logical form with the help of GPT-4 (Achiam et al., 2023) and human feedback. Table 3 shows the formal notation of each form, alongside a sample expression and the total number of unique expressions.

Table 3: Expressions of logical forms

|  | Formal Notation | Sample Expression | No. of Expr. |
|---|---|---|---|
| Basic | $x$ | The claim that $x$ holds true. | 16 |
| Negation | $\neg x$ | The claim that $x$ does not reflect reality. | 15 |
| Conditional | $x \rightarrow y$ | Once we know that $x$, we also know that $y$. | 11 |
| Disjunction | $x \vee y$ | It is a fact that either $x$ or $y$. | 8 |

The variable(s) within each expression is ultimately replaced by randomly selected generic, real-world sentences from GenericsKB-Best (Bhakthavatsalam et al., 2020). The GenericsKB-Best dataset is chosen for its vast collection of simple propositions (1,020,868 sentences) without conditionals, disjunctions, etc. A complete example can be found in Step 2 of Figure 2.

Notably, as shown in Figure 2, the statements are generally factually inaccurate despite being drawn from real-world data. This is intentional. Real-world propositions allow us to generate sentences with diverse grammatical structures that closely emulate human-written arguments. However, factually accurate arguments enable models to bypass deductive reasoning with their prior real-world knowledge, which is experimentally demonstrated in Section 5.1. By using real-world yet factually inaccurate statements, we combine realism and prior knowledge independence.

### 3.3 STEP 3: GENERATE QUERY STATEMENT

The LLM's task is to determine whether the given query statement is true, false, or uncertain based on the premises provided. Using Figure 2 as an example, if we assign the query statement to be the negation of the conclusion, i.e. "It is not true that Japan is in Asia", then the answer is false. If the query statement is the same as the conclusion, then the answer is true. If the query statement is unrelated to the premises, then the answer is uncertain.

### 3.4 DATASET COMPLEXITY

In the context of deductive reasoning datasets, complexity is defined as the variety and comprehensiveness of instances. It can be further divided into two dimensions: natural language complexity and argument complexity. In this section, we highlight the significance of both aspects and how JustLogic compares against other logical reasoning datasets.

Table 4: Statistics of dataset complexity.

|  | Natural Language | | Argument | |
|---|---|---|---|---|
|  | No. of Domains | Vocabulary | Reasoning Depth | Arg. Structures |
| CLUTRR | 1 | 1396 | $1 - \infty$ | $\infty$ |
| ProofWriter | $\times$ | 101 | $1 - \infty$ | $\infty$ |
| LogiQA 2.0 | >10 | 10004 | $\times$ | $\times$ |
| FOLIO | >10 | 4351 | 1 - 7 | 76 |
| JustLogic | >10 | 10557 | $1 - \infty$ | $\infty$ |

**Natural language complexity.** Human language is complex. Statements and arguments of similar meanings can be presented in a variety of ways. Therefore, it is insufficient for models to reason solely with symbols, *e.g.* $x$ and $y$, and basic natural language sentences, *e.g.* "Some birds are yellow."; they must be capable of reasoning with real-world vocabulary and diverse sentence structures to be useful in practical contexts.

We measure natural language complexity with (i) the number of domains, and (ii) vocabulary size. A domain is defined as any topic of interest, such as golf, computers, or traveling; Vocabulary size refers to the number of unique words in the dataset. Appendix D shows text samples of each benchmark to further highlight their linguistic complexity.

As shown in Table 4, existing synthetic datasets have low natural language complexity, while human-written datasets, such as FOLIO and LogiQA 2.0, exhibit significantly higher complexity. This is expected since synthetic datasets translate symbols in formal logic into natural language using limited

templates of sentence structures and vocabulary lists. For example, in ProofWriter, a typical sentence follows the format "All dogs are (not) red.". The linguistic patterns of human-written datasets, in contrast, are bound only by human creativity. Despite being synthetic, JustLogic, achieves natural language complexity on par with manually curated datasets, due to its comprehensive selection of expressions and the use of GenericsKB-Best as the source of sentences.

**Argument complexity.** Argument complexity refers to the diversity of argument structures used in the dataset. A sufficiently high argument complexity is important because humans use a range of argument forms to reason, beyond just conditionals and modus ponens. Moreover, a real-world argument is typically composed of multiple argument forms, due to the inherent complexity of real-life scenarios.

We evaluate a dataset's argument complexity based on two metrics: (i) the range of reasoning depth, and (ii) the number of unique argument structures. The upper limit of both metrics is calculated based on the theoretical maximum without any additional human input, rather than the highest depth used in experiments in existing works. For example, CLUTRR's dataset construction program can generate any number of depths (referred to as relation length in the original paper), despite its experiments only utilizing questions of up to a depth of 10. Thus, its upper limit of depth is infinite.

Table 4 shows that synthetic datasets, such as CLUTRR, ProofWriter, and JustLogic, excel in this area, as there is no upper limit to their reasoning depth and number of argument structures. Manually curated datasets, in contrast, either lack an explicit concept of reasoning depth and argument structures (e.g. LogiQA 2.0), or have a limited selection of both (e.g. FOLIO). While manual datasets require significant human efforts and investment to expand their complexity, synthetic ones can scale trivially.

In summary, JustLogic combines the best aspects of both dataset construction methods, incorporating the argument complexity of synthetic datasets and the natural language complexity of manually curated ones.

## 3.5 FUTURE-PROOFING JUSTLOGIC

As the reasoning abilities of LLMs continue to improve, we expect LLMs to solve the existing JustLogic dataset eventually. As such, its difficulty level must be adjusted to remain relevant as a benchmark for deductive reasoning. We leverage JustLogic's synthetic nature to increase complexity with minimal human input.

Argument complexity can be adjusted by (i) increasing the range of argument depth and (ii) increasing the number of distinct argument forms to $>7$. Natural language complexity can be adjusted by (i) increasing the number of expressions for each logical form and (ii) integrating a more complex knowledge base than GenericsKB. Importantly, these changes are programmatically achievable with minimal man-hours.

## 4 EXPERIMENTAL SETUP

We will first experimentally investigate the influence of prior knowledge on evaluating deductive reasoning with existing benchmarks, using our test for prior knowledge independence. This validates JustLogic's ability to measure deductive reasoning without prior knowledge as a confounder. Next, several SOTA LLMs of various parameter sizes are evaluated using JustLogic. Finally, an in-depth error analysis of the LLMs' results is conducted.

JustLogic contains 7000 instances, with reasoning depths ranging from 1 to 7; each depth has 1000 instances. It is then split into train/validation/test sets, with proportions of 70%/15%/15% or 4900/1050/1050 instances. Train and validation sets facilitate in-context learning and model fine-tuning if required, while the test set is used for evaluation. Note that the number of instances and range of reasoning depths can be easily adjusted using JustLogic's open-source dataset generation program.

### 4.1 PRIOR KNOWLEDGE INDEPENDENCE TEST

The task for deductive reasoning benchmarks is typically framed as $CQO \rightarrow A$: Given a context $C$, consisting of $n$ premises ($P = \{p_1, p_2, ..., p_n\}$), a question $Q$, and $m$ answer options ($O =$

$\{o_1, o_2, ..., o_m\}$), determine the correct answer $A$. To assess the influence of prior knowledge on determining answer $A$, the prior knowledge independence test is framed as $QO \rightarrow A$. No context $C$ is provided, and the prompt instructs the LLM to answer the question based on prior knowledge alone. An example is provided in Appendix A.

If prior knowledge is not useful, the LLM should be unable to answer question $Q$ without $C$, and the accuracy for the prior knowledge independence test should approximate random probability $\frac{1}{m}$. Benchmarks exhibiting such accuracies are deemed prior knowledge independent.

Any LLM capable of using prior knowledge can be used for this test. However, models with larger parameter sizes, and thus more extensive prior knowledge, are more likely to exhibit notable differences in accuracies. For our experiment, we use GPT-4. The test is conducted on both JustLogic and existing benchmarks, including CLUTRR, ProofWriter, LogiQA 2.0, and FOLIO.

## 4.2 Evaluation of LLMs' Deductive Reasoning

Our task follows the conventional formulation: $CQO \rightarrow A$. Question $Q$ is "Is the statement $S$ true, false, or uncertain?", followed by the query statement, as shown in Figure 1; there are 3 answer options, where $O = \{\text{true}, \text{false}, \text{uncertain}\}$. All prompts begin with a preamble, which includes (i) the requirements of the task at hand, (ii) a list of argument forms in propositional logic, and (iii) the available answer options.

We evaluated various models of different sizes, including Llama3-8B (Dubey et al., 2024), Llama3-70B, GPT-4, GPT-4o, and OpenAI o1-preview (OpenAI, 2024b). Given that prompt quality significantly impacts LLM accuracy, a range of prompting techniques are tested: zero-shot, few-shot, and chain-of-thought (CoT) (Wei et al., 2022). OpenAI o1-preview had strict rate limits at the time of writing since it was released less than a month prior to manuscript submission. As such, 42 random samples in the test set are used for OpenAI o1-preview. To ensure fairness, the selected subset has the same proportion of reasoning depth and classes (True, False, and Uncertain) as the entire test set. Further implementation details are provided in Appendix B.

We also measured human performance. 18 anonymous participants, recruited from Amazon Mechanical Turk [2], are given a random subset of questions. This is because deductive reasoning questions, especially those at high reasoning depths, are cognitively demanding and time-consuming; it is impractical to expect humans to complete 1050 questions. To ensure fairness, both models and participants are provided similar prompts and are given the same proportion of each reasoning depth.

Finally, we perform an error analysis of the results from the aforementioned experiments, specifically examining the heterogeneous effects of argument form and reasoning depth on model accuracy. Accuracy for each argument form is only measured using questions with a reasoning depth of 1 since those with a depth of $>1$ typically have $>1$ argument forms.

# 5 Results

## 5.1 Prior Knowledge Independence Test

The results of JustLogic and four other benchmarks are shown in Table 5; note that lower accuracy relative to the benchmark's random probability indicates that prior knowledge is more detrimental to answering the question, thereby demonstrating that the benchmark is more prior knowledge independent. The accuracies of CLUTRR and ProofWriter are close to random probability, while those of LogiQA 2.0 and FOLIO are nontrivially higher. This is because the former are synthetic datasets, while the latter are manually curated. When a question is code-generated, it generally bears no correlation with reality, e.g. "Is it true, false, or uncertain that Gary is not red." from ProofWriter and "How is Anna related to Katherine in the family?" from CLUTRR. Such questions are only answerable by reasoning over the context $C$. LogiQA 2.0 and FOLIO, on the other hand, often contain questions that are answerable without the context provided. For example, "The United States won the most medals in the last summer Olympic games." from FOLIO can be accurately answered by LLMs trained on sufficiently recent general knowledge datasets. We hypothesize that this is an

---

[2]Website: https://www.mturk.com/

unintentional consequence of the human bias to align the question's truth value with reality. While human curation enhances the question's realism, it compromises the test for deductive reasoning.

Table 5: Results of Prior Knowledge Independence Test. **The closer to Random Prob., the better.**

|  | Accuracy (%) | Random Prob. (%) |
| --- | --- | --- |
| CLUTRR (Sinha et al., 2019) | 8.3 | 6.25 |
| ProofWriter (Tafjord et al., 2020) | 37.0 | 33.3 |
| LogiQA 2.0 (Liu et al., 2023a) | 52.1 | 25.0 |
| FOLIO (Han et al., 2022) | 40.0 | 33.3 |
| JustLogic | **33.7** | 33.3 |

The JustLogic benchmark's accuracy (33.7%) is the closest to random probability (33.3%) compared to other benchmarks, including synthetic ones. The reason for this is twofold: first, JustLogic is also a synthetic dataset, which eliminates the human bias present in manually curated datasets. Second, while JustLogic uses real-world statements, their truth value is nonetheless randomly determined. For example, the statement "doors are solids" is factually true. However, by deducing from the paragraph, the correct answer is "False". Thus, using prior knowledge for many questions is not only unhelpful but also meaningfully decreases accuracy.

## 5.2 EVALUATION OF LLMs' DEDUCTIVE REASONING

As shown in Table 6, the best-performing model by a large margin is OpenAI o1-preview with an accuracy of 81.0%. The second and third-best models, GPT-4o and Llama3-70B, achieved 65.6% and 64.6% respectively. Models with larger parameter sizes generally perform better than smaller models, assuming the same prompting methods are used. For example, zero-shot Llama3-8B achieved an accuracy of 49.8%, while zero-shot Llama3-70B achieved an accuracy of 53.1%. However, larger model sizes offer diminishing returns, shown by the accuracy gain of just 1.0% from Llama3-70B to GPT-4o, with both using CoT prompting.

Moreover, the improvements offered by increasing model size pale in comparison to those offered by better prompting methods. Using chain-of-thought prompting, Llama3-8B achieved higher performance (57.8%) than zero-shot Llama3-70B (53.1%). This appears to explain the significant accuracy gap of 15.4% between OpenAI o1-preview and its non-reasoning-focused counterpart, GPT-4o. OpenAI o1-preview is trained to reason with chain-of-thought prompts using a 'reinforcement learning algorithm' (OpenAI, 2024a). We hypothesize that the use of reinforcement learning on CoT prompting further enhances the deductive reasoning capabilities offered by CoT prompting alone.

Table 6: Model and Human Evaluation Results

| Model | Prompting Method | Accuracy (%) |
| --- | --- | --- |
| Random Probability |  | 33.3 |
| Llama3-8B | Zero-shot | 49.8 |
| Llama3-8B | Few-shot | 41.8 |
| Llama3-8B | CoT | 57.8 |
| Llama3-70B | Zero-shot | 53.1 |
| Llama3-70B | Few-shot | 57.8 |
| Llama3-70B | CoT | 64.6 |
| GPT-4 | CoT | 59.2 |
| GPT-4o | CoT | 65.6 |
| OpenAI o1-preview | CoT | 81.0 |
| Human Average |  | 73.0 |
| Human Ceiling |  | 100.0 |

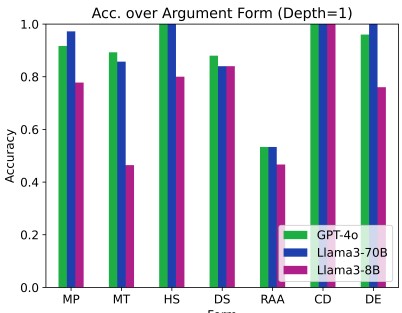 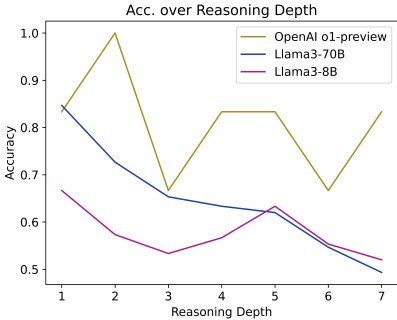

Figure 3: How argument form[3] and reasoning depth affects accuracy for various models

Human performance (73.0%) is significantly higher than all models besides OpenAI o1-preview, while the human ceiling (100.0%) outperforms all models. The non-trivial gap between the human ceiling and the best-performing model (81.0%) shows that models still have significant room for improvement. Moreover, we believe actual human performance might be higher than 73.0%. Given the long paragraphs of questions with high reasoning depth, participants may have predicted answers by briefly scanning the paragraph, rather than carefully deducing based on all available premises. This is supported by the suspiciously short time taken to complete the questions of some participants.

## 5.3 ERROR ANALYSIS

Figure 3 illustrates the model accuracy by argument form (left) and by reasoning depth (right). It shows the statistics for (i) Llama3-8B with CoT prompting, chosen for its superior performance amongst smaller LLMs, (ii) Llama3-70B with CoT prompting, chosen due to its superior performance amongst medium-sized LLMs, and (iii) OpenAI o1-preview, chosen for its overall highest performance. Note that OpenAI o1-preview is excluded from the argument form analysis due to insufficient samples; GPT-4o is displayed instead for a more comprehensive comparison.

The accuracies of some argument forms are evidently better than others. For example, hypothetical syllogism and constructive dilemma achieve considerably higher performance than modus tollens and reductio ad absurdum. We hypothesize that these forms appear less frequently in the models' training data. With less exposure to them, models may overlook these argument forms in favor of more common ones during deductive reasoning, owing to the probabilistic nature of neural networks (Fahlman & Hinton, 1987).

As for reasoning depth, model accuracies generally decrease as depth increases, consistent with expectations that accuracy declines as the complexity of questions increases. Interestingly, Llama3-70B performs comparably to OpenAI o1-preview for instances with a depth of 1, but Llama3-70B sees a sharp decline in performance once depth is increased, while OpenAI o1-preview only sees a moderate decline; OpenAI o1-previews' superior performance is a result of better reasoning at higher reasoning depths. This seems to suggest OpenAI o1-preview's CoT prompting supports deeper and longer lines of reasoning, which is crucial for deductive reasoning. Fluctuations on all three trendlines are likely due to small sample sizes: OpenAI o1-preview has 6 samples per depth, while the other 2 have 150. We expect the trend to be more explicit with a larger number of samples.

## 6 CONCLUSION

Deductive reasoning is one of the key challenges in LLM research. In response to the lack of reliable benchmarks, we present JustLogic, a natural language deductive reasoning dataset that is (i) highly complex, (ii) prior knowledge independent, and (iii) capable of in-depth error analysis. These qualities are enabled by JustLogic's dataset construction method: argument structures are synthetically generated, and natural language is programmatically incorporated via expression templates and a knowledge base. We empirically justify JustLogic's merits. Moreover, most LLMs underperform the human average and all LLMs significantly underperform the human ceiling. We demonstrate that JustLogic is a highly challenging, future-proof benchmark that is reliable and insightful for evaluating logical reasoning in LLMs.

---

[3]MP = Modus Ponens, MT = Modus Tollens, HS = Hypothetical Syllogism, DS = Disjunctive Syllogism, RAA = Reductio Ad Absurdum, CD = Constructive Dilemma, DE = Disjunction Elimination

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

## A  PRIOR KNOWLEDGE INDEPENDENCE TEST

A sample prompt for the prior knowledge independence test, based on the example in Figure 1, is shown below in Figure 4. Note that the answer options vary depending on the benchmark. For example, the options for LogiQA are A, B, C, and D, while those of CLUTRR are 16 possible family relations.

> **Instructions:**
> - Use the knowledge you currently have to answer as accurately as possible.
> - You have 3 answer options: True, False, and Uncertain.
> - There should be roughly an equal proportion of each option.
> - *Add 5-10 examples here*
>
> **Question:** Is the following statement true, false, or uncertain?
> **Statement:** Doors are solids.
> **Answer:** True.

Figure 4: Example of a prior knowledge independence test prompt

## B  EXPERIMENT IMPLEMENTATION DETAILS

The hyperparameters for the Llama3 models are decided largely based on the recommendations in the original paper Dubey et al. (2024), which are as follows: temperature of 0.6, top p of 0.9, presence penalty of 1.15, length penalty of 1.

With regards to prompting methods, 3-shot prompting is chosen for few-shot experiments because it produces the highest accuracies compared to 6 and 9-shot. Chain-of-thought prompts also contain three examples. In the interest of fairness, all prompting techniques contain similar instructions, which are as follows:

> You are given a paragraph of facts/premises, followed by a statement. Perform logical reasoning with propositional logic on the paragraph to determine the truth value of the statement.
>
> Here is the list of argument forms:
> - Modus Ponens
> - Modus Tollens
> - Hypothetical Syllogism
> - Disjunctive Syllogism
> - Reductio ad absurdum
> - Constructive Dilemma
> - Disjunction Elimination
>
> You must answer with either one of the 3 options:
> - TRUE: When the premises in the paragraph lead to the statement
> - FALSE: When the premises in the paragraph directly contradict the statement
> - UNCERTAIN: When the premises in the paragraph neither support nor contradict the statement
>
> Do not use your prior knowledge; your answer must be solely determined by the information within the paragraph. Assume that all premises in the paragraph are true.
>
> Question: Is the statement true, false, or uncertain?

## C IMPACT OF FACTUAL ACCURACY ON MODEL PERFORMANCE

Given that JustLogic randomly chooses sentences from GenericsKB to add to each instance's argument structure, the final conclusion may be factually accurate or inaccurate in the real world. For example, if the conclusion is "It is not true that Japan is in Asia.", then the conclusion is factually inaccurate. There is therefore a concern that models underperform due to confusion arising from factually inaccurate conclusions. Moreover, since some conclusions are factually accurate, such instances may exhibit artificially high performance.

To study these concerns, we conduct the following empirical study. If the above concerns are true, we expect factually inaccurate conclusions to perform worse than factually accurate ones. Because all GenericsKB sentences are factually accurate, we can straightforwardly deduce each conclusion's factual accuracy. For example, $x \lor y$ is factually accurate while $\neg x$ is not.

Figure 5 shows the comparison of accuracies for five models: OpenAI o1-preview, GPT-4o, GPT-4, Llama3-70B, Llama3-8B; the left represents when reasoning depth is 1 and the right represents when depth is 7 or less.

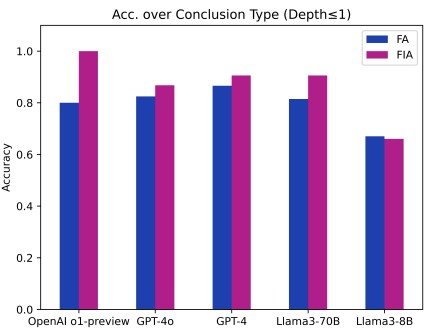 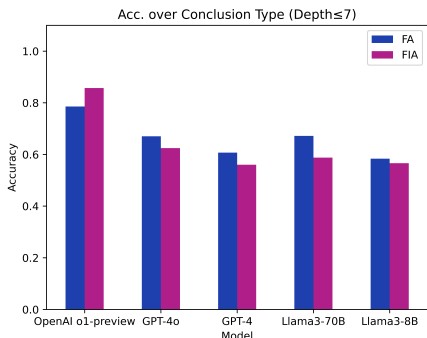

Figure 5: How factual accuracy of conclusions affects model accuracy

These results reject the hypothesis that factually inaccurate conclusions perform worse than factually accurate ones; there is no consistent trend between both conclusion types. In fact, when depth=1, factually inaccurate conclusions exhibit higher performance! This trend is somewhat reversed when depth is 7 or less, but OpenAI o1-preview is a notable exception.

There are two reasons for these results. First, our prompt explicitly instructs models to answer the question only using the paragraph provided and without using prior knowledge. The full prompt is shown in Appendix B. Moreover, in few-shot prompts, the examples provided include conclusions where their factual accuracy does not match the correct answer. These measures encourage models to ignore prior knowledge and answer questions without considering the factual accuracy of conclusions in the real world.

Second, how LLMs treat factual accuracy when reasoning deductively depends on the LLM's training: specifically, the model's ability to follow prompt instructions to ignore prior knowledge. For example, OpenAI o1-preview biases towards factually inaccurate conclusions when deductively reasoning, while Llama3-8B exhibits no difference in performance. Should an LLM exhibit significant differences in performance between factually accurate and inaccurate conclusions, it suggests the LLM has room for improvement in instruction following.

Importantly, the ability to deduce whether premises lead to a conclusion without using prior knowledge is a fundamental human skill: we use it to evaluate whether a debater's speech or journalist's article supports their position. The inclusion of both factually accurate and inaccurate instances in JustLogic is a feature, not a bug.

## D SAMPLE TEXTS FROM DEDUCTIVE REASONING BENCHMARKS

Beyond metrics like vocabulary size and number of domains, the degree of natural language complexity can be straightforwardly determined by manually inspecting the linguistic patterns of a given

Table 7: Sample texts from various deductive reasoning benchmarks

| Benchmark | Sample Text |
|---|---|
| CLUTRR (Sinha et al., 2019) | Lorraine and her brother Kevin went to see a movie. Clarence took his granddaughter Lorraine to the movies and they enjoyed themselves. |
| ProofWriter (Tafjord et al., 2020) | The bald eagle is not rough. The bear does not need the bald eagle. The dog needs the bear. If someone is rough then they chase the bald eagle. If someone needs the bear then they are not blue... |
| ProntoQA-OOD (Saparov et al., 2024) | Lempuses are bitter. Every lempus is a lorpus. Brimpuses are vumpuses. Tumpuses are impuses. Each impus is not hot. Every numpus is a sterpus. Each shumpus is brown. Sterpuses are fast. Every vumpus is not small... |
| LogiQA 2.0 (Liu et al., 2023a) | In the past 10 years, the sales of personal notebook computers of a computer company have continued to grow, but the growth rate is lower than the growth rate of the company's total sales of all products. |
| FOLIO (Han et al., 2022) | All people who regularly drink coffee are dependent on caffeine. People regularly drink coffee, or they don't want to be addicted to caffeine, or both. No one who doesn't want to be addicted to caffeine is unaware that caffeine is a drug... |
| JustLogic | Either one or both of these statements are true: big head is another sudden death disease which occurs primarily in feedlot cattle, or some energy is transferred by bulbs. The notion that 'big head is another sudden death disease which occurs primarily in feedlot cattle' is untrue. |

benchmark. Table 7 shows sample texts from CLUTRR, ProofWriter, ProntoQA-OOD, LogiQA 2.0, FOLIO, and JustLogic.

Evidently, JustLogic exhibits significantly greater natural language complexity than CLUTRR, ProofWriter, and ProntoQA-OOD, because the latter benchmarks programmatically generate every sentence, while JustLogic extracts its sentences from GenericsKB, a natural language text database. Thus, CLUTRR, ProofWriter, and ProntoQA-OOD rely on a limited number of grammar templates, reducing their linguistic complexity. JustLogic exhibits similar levels of complexity to FOLIO. LogiQA 2.0 is more complex because it is human-curated and not backed by a formal logic system (unlike how JustLogic is backed by propositional logic). Without a formal logic system, LogiQA 2.0's argument complexity suffers, as shown in Table 4, which compromises its ability to evaluate deductive reasoning in LLMs.

## E    FUTURE WORKS

While JustLogic already achieves higher or similar natural language complexity to existing deductive reasoning benchmarks, as shown in Section 3.4, linguistic complexity can be further enhanced to emulate human-written prose, e.g. news articles and fiction stories. Notably, LLMs can be introduced in Step 2 of JustLogic's dataset construction process, whereby instead of randomly selecting sentences from GenericsKB, an LLM can generate fictional statements and scenarios, e.g. "John's favorite food is hamburgers.". While LLM generation has been successful in datasets involving inductive reasoning and commonsense knowledge, e.g. MuSR (Sprague et al., 2023), it is currently too unreliable for deductive reasoning due to several common mistakes, e.g. ignoring instructions, hallucination, and invalid logic. Nonetheless, as LLMs become more reliable, LLM generation is a promising approach worthy of further exploration.

Error analysis using JustLogic can also be further explored. Interesting research questions include: Are models able to use argument forms appropriately? At which step of the argument chain does the model usually fail? What are the most common reasons for failure? These insights may be useful for fine-tuning models for logical reasoning tasks (Liu et al., 2023b) and model guidance (Beurer-Kellner et al., 2024).

JustLogic has a single question type, i.e. based on the context, determine whether a given statement is true, false, or uncertain. However, there are many other question types relevant to logical reasoning, such as multiple-choice questions, identifying missing premises in arguments, identifying logical fallacies in arguments, and natural language sentence to formal logic translation. Liu et al. (2023b) provides a comprehensive taxonomy. JustLogic's program can be adapted to accommodate each question type while maintaining its key advantages. By measuring deductive reasoning across multiple modalities using a single dataset construction method, JustLogic can provide more comprehensive and controlled evaluations and error analysis.

