# OpenReview forum: "JustLogic: A benchmark for natural language deductive reasoning"
_ICLR.cc/2025/Conference — Submitted to ICLR 2025_

### Official Review · Reviewer_LXqP · 2024-11-03

**Soundness:** 3
**Presentation:** 3
**Contribution:** 2
**Rating:** 5
**Confidence:** 4

**Summary:**

The paper introduces a synthetic benchmark JustLogic to assess the deductive reasoning capabilities of LLMs. Key contributions include
* Benchmark Design: JustLogic is a benchmark specifically for evaluating LLMs' logical reasoning skills. It minimizes the influence of prior knowledge, ensuring that models rely on logical deduction alone rather than external knowledge.
* Evaluation of Existing Models: JustLogic's experiments reveal that current LLMs, such as GPT-4 and Llama3 variants, perform significantly below human accuracy on deductive tasks, indicating substantial room for improvement in LLM reasoning capabilities.
* Error Analysis: JustLogic allows detailed analysis of model errors, exploring how factors like argument form and reasoning depth affect model accuracy, thus offering insights into specific areas of reasoning weakness in LLMs.

**Strengths:**

* Originality: This paper contributes a well-defined deductive reasoning benchmark addressing limitations in prior benchmarks by focusing on memorization-aware design and error analysis. The originality lies in synthesizing complex deductive tasks without relying on real-world facts, thus eliminating the influence of prior knowledge, which is usually ignored in existing benchmarks.

* Quality : JustLogic’s design is grounded in logical principles, using well-established argument forms (e.g., modus ponens) to construct queries. It provides a comprehensive evaluation across multiple LLMs and include human baselines, showing the benchmark’s realism and effectiveness. The extensive error analysis further strengthens the paper.

* Clarity: The paper is clearly written and organized into motivation, dataset construction, and evaluation.

* Significance: This work contributes to LLM reasoning evaluation, as it introduces a benchmark that better reflects true deductive reasoning without overly reliance on memorized facts.

**Weaknesses:**

1. “it is important to note that, despite being drawn from real-world data, the statements are generally factually inaccurate. This is intentional.” “By using real-world yet factually inaccurate statements, we combine realism and context-independence.”

It does not make much sense to “intentionally” choose factually inaccurate statements. The authors claim to study reasoning from a perspective of “eliminating the influence of prior knowledge and ensuring context independence” (L93). However, the effect of prior knowledge is exerted from both the positive and negative sides, i.e. certain knowledge could align with or disagree with a given query. For example, if a model is already pretty sure about the fact that “Japan is in Asia”, then this degree of certainty would likely to affect the reasoning about the question that leads to “Japan is not in Asia”, so the influence of prior knowledge is not “eliminated”. In this sense, the topic would be counterfactual reasoning, which has been studied by a few previous work [1]. A more reasonable principle of choosing the statements would be complete irrelevance of real-world knowledge, e.g. “My cat Fluffy likes paper bags”.

2. “The variable(s) within each expression is ultimately replaced by randomly selected generic, realworld sentences from GenericsKB-Best”

It would be better to include more details about the GenericsKB-Best dataset. One of the concerns would be: Is it assured that any two statements in the dataset are semantically independent with each other? For example, if statement A is “Japan is in Asia”, while statement B is “Japan is not in Asia”, then “A and B” will actually reduce to False. This problem would be more pronounced when the “argument complexity” is scaled up, as there would be more statements involved.

3. Error analysis is somewhat superficial. It is not surprising to see a decrease in performance as the reasoning depth increases. A more interesting investigation would be regarding at which step the model starts to fail and why, as it would be a direct guidance of steering the model towards the correct reasoning path and avoid straying from the subject.

4. It would be better to add another section to prove the robustness of the benchmark by finetuning a model on the synthetic deductive data, showing whether the benchmark can be hacked by training on test data.


[1] Wu, Zhaofeng, et al. "Reasoning or reciting? exploring the capabilities and limitations of language models through counterfactual tasks." arXiv preprint arXiv:2307.02477 (2023).

**Questions:**

See weaknesses.

---

> ### Author Response · Authors · 2024-11-17
> **Response to Reviewer LXqP (Part 1)**
>
> Thank you for your detailed and constructive reviews! We’re glad that you appreciate how JustLogic addresses the limitations of existing deductive reasoning benchmarks.
>
> > W1: It does not make much sense to “intentionally” choose factually inaccurate statements…For example, if a model is already pretty sure about the fact that “Japan is in Asia”, then this degree of certainty would likely to affect the reasoning about the question that leads to “Japan is not in Asia”, so the influence of prior knowledge is not “eliminated”...A more reasonable principle of choosing the statements would be complete irrelevance of real-world knowledge, e.g. “My cat Fluffy likes paper bags”.
>
> You are indeed right that the effects of prior knowledge can be on both positive and negative sides. However, **it is not the case that if a conclusion is factually inaccurate, models may use prior knowledge and therefore lead to artificially low accuracies.**
>
> First, our prompt explicitly instructs models to answer the question only using the paragraph provided, and not to use prior knowledge (Appendix B). Moreover, in few-shot prompts, the examples provided include conclusions where their factual accuracy does not match the correct answer. These measures encourage models to ignore prior knowledge even when conclusions are factually accurate.
>
> Second, we conducted an additional empirical study showing that the factual accuracy of conclusions has no clear impact on model accuracy. The full methodology and results are shown in Appendix C. If prior knowledge interferes with the deduction of factually inaccurate conclusions, then such questions have lower accuracies.
>
> However, our results show that when reasoning depth is 1, factually inaccurate conclusions in fact exhibit higher performance than factually accurate ones! When reasoning depth is 7 or less, factually inaccurate conclusions exhibit lower performance for Llama3-8B, Llama3-70B, GPT-4, and GPT-4o, but the opposite is true for OpenAI o1-preview. Moreover, the difference in accuracy is typically marginal. Further explanations of these observations can be found in Appendix C.
>
> **These results show that the factual accuracy of conclusions has no clear or meaningful impact on accuracy, which suggests prior knowledge independence is upheld regardless. Therefore, factually inaccurate conclusions do not confuse models due to contradicting prior knowledge.**
>
> With regard to using statements irrelevant to real-world knowledge, the existing literature suggests that this approach exhibits significant flaws. In order to generate fictional stories or arguments, every sentence must be constructed from scratch, rather than derived from a natural language text database like GenericsKB. There are currently two ways of accomplishing this: (i) programmatically and (ii) using an LLM.
>
> ProofWriter [1] programmatically generates fictional sentences using limited grammar. The following is a sample text: “The bald eagle is not rough. The bear does not need the bald eagle. The dog needs the bear…”. More examples from CLUTRR and ProntoQA can be found in Appendix D. Evidently, programmatic generations significantly compromise natural language complexity. Sentences no longer represent real-world scenarios and domains. Such datasets are significantly less realistic than JustLogic.
>
> MuSR [2] uses GPT-4 to generate fictional stories. While it does have high natural language complexity, the stories sometimes contain mistakes, e.g. ignoring instructions, hallucination, and invalid logic. MuSR’s tasks also test for inductive reasoning and commonsense knowledge, so minor inaccuracies in the stories do not affect the viability of the dataset. However, JustLogic solely tests for deductive reasoning, so any imprecision in the premises will make deductive reasoning impossible, therefore rendering the instance useless. If GPT-4 is used to generate JustLogic’s premises, the dataset will no longer be 100% reliable. Therefore, MuSR's method of using GPT-4 is not viable for JustLogic's purpose.
>
> **To summarize, the intentional use of factually inaccurate conclusions does not cause confusion in models, therefore making JustLogic’s approach justified for ensuring prior knowledge independence. Moreover, JustLogic’s approach is superior to alternative approaches due to their significant flaws.**
>
> > W2: One of the concerns would be: Is it assured that any two statements in the dataset are semantically independent with each other?
>
> Regarding the example of contradictory statements, this is not possible because all sentences in GenericsKB are true.
>
> It is possible that some sentences are semantically related. However, the chances are extremely small, considering there are 3.4M unique sentences across 129,987 topics.
>
> To further mitigate this, we will edit the JustLogic program such that every GenericsKB sentence in a paragraph belongs to a different topic, thus further reducing the chance of accidental semantic dependence.

---

> ### Author Response · Authors · 2024-11-17
> **Response to Reviewer LXqP (Part 2)**
>
> > W3: Error analysis is somewhat superficial…A more interesting investigation would be regarding at which step the model starts to fail and why, as it would be a direct guidance of steering the model towards the correct reasoning path and avoid straying from the subject.
>
> We have considered conducting such an investigation. However, this requires considerable effort because different models structure their CoT responses differently. In fact, different questions within the same LLM can result in vastly different forms of CoT. Thus, every response must be manually and individually parsed to identify where exactly the model failed.
>
> Moreover, based on our examination of several LLM responses, there is a wide and complex array of potential errors, including hallucinated premises, not using relevant premises, incorrect usage of argument forms, and even responses that are simply incoherent. Such an analysis requires careful and extensive study, which is beyond the scope of this paper.
>
> The error analysis (Section 5.3), where we study how reasoning depth and argument form affect accuracy for various models, is the first step toward a more thorough investigation of where models fail and why.
>
> We agree that this is an interesting and promising direction, which we intend to explore in the future. A discussion on this has been added to the Future Works section (Appendix E). Thank you for the suggestion!
>
> > W4: It would be better to add another section to prove the robustness of the benchmark by finetuning a model on the synthetic deductive data, showing whether the benchmark can be hacked by training on test data.
>
> As observed in other deductive reasoning datasets, finetuning on the same dataset always leads to some degree of ‘hackability’. Fine-tuned models exhibit significant performance improvements. In FOLIO [3], a human-curated dataset, the fine-tuned Flan-T5-Large (783M parameters) outperforms GPT-4. In CLUTRR [4], a synthetic dataset, models were finetuned on a dataset with depth=2 and 3. Graph Attention Network achieved near-perfect performance on these depths, but performed significantly worse as depth is increased.
>
> Therefore, we expect models to perform better after being finetuned on the JustLogic dataset. In line with other deductive reasoning benchmarks, this does not challenge the robustness of JustLogic.
>
> Moreover, two features of JustLogic reduce its hackability. First, JustLogic’s reasoning depth can be increased. Just like CLUTRR, if the model is finetuned on depth=2 and 3, then the model can be tested on depths > 3, in the event that the model memorizes all the possible permutations of argument forms at lower depths.
>
> Second, JustLogic’s training and test data can have completely different linguistic structures. Specifically, GenericsKB sentences that appear in the train data will not appear in the test data. Moreover, the expressions of each logical form (Table 3) used in the train data will not be used in the test data. This can be trivially implemented and verified using JustLogic’s program.
>
>
> [1] Oyvind Tafjord, Bhavana Dalvi, and Peter Clark. 2021. ProofWriter: Generating Implications, Proofs, and Abductive Statements over Natural Language. In Findings of the Association for Computational Linguistics: ACL-IJCNLP 2021, pages 3621–3634, Online. Association for Computational Linguistics.
>
> [2] Sprague, Z., Ye, X., Bostrom, K., Chaudhuri, S. and Durrett, G. 2023. Musr: Testing the limits of chain-of-thought with multistep soft reasoning. arXiv preprint arXiv:2310.16049.
>
> [3] Simeng Han, Hailey Schoelkopf, Yilun Zhao, Zhenting Qi, Martin Riddell, Luke Benson, Lucy
> Sun, Ekaterina Zubova, Yujie Qiao, Matthew Burtell, David Peng, Jonathan Fan, Yixin Liu, Brian
> Wong, Malcolm Sailor, Ansong Ni, Linyong Nan, Jungo Kasai, Tao Yu, Rui Zhang, Shafiq Joty,
> Alexander R. Fabbri, Wojciech Kryscinski, Xi Victoria Lin, Caiming Xiong, and Dragomir Radev.
> Folio: Natural language reasoning with first-order logic. arXiv preprint arXiv:2209.00840, 2022.
> URL https://arxiv.org/abs/2209.00840.
>
> [4] Koustuv Sinha, Shagun Sodhani, Jin Dong, Joelle Pineau, and William L Hamilton. Clutrr: A
> diagnostic benchmark for inductive reasoning from text. arXiv preprint arXiv:1908.06177, 2019.

---

> ### Comment · Reviewer_LXqP · 2024-12-02
> **Response to Rebuttal**
>
> Thank you for your feedback. I appreciate your efforts, but your feedback does not fully convince me.
>
> "You tell te model not to use prior knowledge" does not mean it genuinely does not use prior knowledge. For example, in system prompt I could say "do not say any words that might be harmful to human society", but I can still easily trigger the model to say something like this. Also, "However, the chances are extremely small, considering there are 3.4M unique sentences across 129,987 topics.", I don't think 3M / 129k is a small number. You need to further justify this in your newest draft if you have not done so yet.
>
> " Thus, every response must be manually and individually parsed to identify where exactly the model failed." I don't think this should be left to future work. You should at least give an example of one popular model to show how your proposed task is challenging to it. Your error analysis is interesting but not somewhat lack of depth.

---

> > ### Author Response · Authors · 2024-12-03
> > **Response to Reviewer LXqP**
> >
> > Thank you for taking the time to consider our rebuttal and engage in this discussion.
> >
> > > "You tell te model not to use prior knowledge" does not mean it genuinely does not use prior knowledge.
> >
> > Indeed, prompts do not guarantee that LLMs would not nonetheless attempt to use prior knowledge. However, our newly added empirical experiment (Appendix C) shows that LLMs do not have this problem. If prior knowledge continues to be used, we should expect questions with factually inaccurate conclusions to have lower accuracies. However, this is not the case: **it has no clear or meaningful impact on accuracy.**
> >
> > > Also, "However, the chances are extremely small, considering there are 3.4M unique sentences across 129,987 topics.", I don't think 3M / 129k is a small number. You need to further justify this in your newest draft if you have not done so yet.
> >
> > 3M / 129k = ~26 sentences per topic. This is rather small, considering the sheer number of topics available. More importantly, given that all GenericsKB sentences are factually accurate, they will not contradict each other and cause confusion in LLMs.

---

### Official Review · Reviewer_3iPx · 2024-11-03

**Soundness:** 2
**Presentation:** 3
**Contribution:** 2
**Rating:** 5
**Confidence:** 5

**Summary:**

This paper describes a procedure for synthesizing data to test the logical reasoning abilities of language models.
The proposed method consists of:
1. Constructing random propositional logic arguments
2. Substituting natural language statements drawn at random from GenericsKB for the propositional variables
3. Sampling a truth value for the example (from True/False/Uncertain) and generating a query statement accordingly (either the conclusion of the argument, its negation, or a random related statement from GenericsKB, respectively)

The authors use this procedure to construct a challenge dataset, JustLogic, and evaluate several LLMs on it, in addition to conducting a human baseline study on MTurk. The human average is 73%, while OpenAI o1-preview gets 81%. The authors state that the 'human ceiling' is 100%, but do not clarify whether this is the accuracy of the best-performing participant, or just implying that every question was answered correctly by at least one participant.

**Strengths:**

The authors rightly state that conflating prior knowledge and inferred knowledge is an issue with certain ubiquitous benchmarks, and causes problems when they are used to make claims about reasoning. It is helpful to have more challenging datasets that are controlled to ensure performance reflects models' ability to reason from provided context, rather than other extraneous features.

**Weaknesses:**

Two of the properties of JustLogic claimed as virtues by the authors are somewhat dubious:
- W1: **The truth value of a statement is unrelated to its truth value in the real world** (termed "context-independence" by the authors). This seems to me to be mostly a source of confusion for both models and human labelers, as the dataset tacitly overrides the truth values of very basic facts. The value of "context-independence" is ensuring that answers can't be guessed based on questions in the absence of premises (as the authors test in 5.1). However, this can be accomplished without contradicting world knowledge - see for example MuSR (Sprague et al., 2024), in which premises involve world knowledge but answer options are controlled so that correctness solely reflects argument structure.
- W2: **High "natural language complexity", by which the authors mean vocabulary size.** I think this is a straightforward overclaim: the linguistic complexity of the task does not increase just because propositional variables have been substituted for longer or more diverse spans. A nontrivial notion of natural language complexity would imply a higher degree of complexity in the _relationship between_ the natural language and the logical argument structure, i.e. comprehending a wider range of linguistic structures would be required in order to recover the logical structure. In my opinion this would also imply moving beyond propositional logic, as natural language largely revolves around predicate-argument structure.

Additionally:
- W3: The 'future-proofness' of the dataset is called into question by the fact that o1-preview outperforms the human average. Additional results showing that the dataset's complexity knobs can be turned up to counteract this would mitigate this weakness. As it stands, the authors claim that the dataset can be scaled to match stronger models but do not actually demonstrate this for the strongest model.

**Questions:**

**Suggestion:** It would be helpful for clarity if you could use a term other than 'context-independence' to describe the property of answer correctness not being influenced by prior knowledge, as 'context-independent' is ambiguous and somewhat counterintuitive: in fact, answers in JustLogic are _only_ sensitive to the question and context (JustLogic accuracy goes to random chance when C is omitted from the CQO tuple in 5.1). The clarity of the paper's arguments would be improved by avoiding the use of 'context' to refer to both world knowledge and question premises.

**Q1:** How is the human ceiling computed?

---

> ### Author Response · Authors · 2024-11-17
> **Response to Reviewer 3iPx (Part 1)**
>
> Thank you for your detailed and constructive reviews! We’re glad that you appreciate the significance of the flaws in current reasoning benchmarks and how JustLogic addresses them.
>
> > W1: …mostly a source of confusion for both models and human labelers…However, this can be accomplished without contradicting world knowledge - see for example MuSR (Sprague et al., 2024)...
>
> **We conducted an additional empirical study showing that overriding the truth value of statements does not lead to confusion for both models and human labelers.**
>
> For models, we conducted an empirical study during the discussion period that proves statements whose truth values are unrelated to the real world do not lead to confusion. The full methodology and results are shown in Appendix C. If such statements lead to confusion, we should find that they exhibit lower accuracies.
>
> However, our results show that when reasoning depth is 1, factually inaccurate conclusions in fact exhibit higher performance than factually accurate ones! When reasoning depth is 7 or less, factually inaccurate conclusions exhibit lower performance for Llama3-8B, Llama3-70B, GPT-4, and GPT-4o, but the opposite is true for OpenAI o1-preview. Moreover, the difference in accuracy is typically marginal.
>
> These results show that the factual accuracy of conclusions has no clear or meaningful impact on accuracy, which suggests such conclusions do not lead to confusion.
>
> For humans, we find that participants of our human evaluation are able to reason about the truth value of the statement while ignoring its truth value in the real world. First, our instructions explicitly ask participants to “assume everything in the paragraph is true” and to “only use information found in the paragraph”. We also anecdotally confirmed with several participants that our instructions were clear. Second, the perfect human ceiling performance of 100% and strong average human performance of 73.0% indicate that participants are not confused by such statements.
>
> Third, the ability to deductively reason without drawing on the truth value in the real world is a fundamental human skill that is commonly used in daily life. For example, to evaluate whether a debater’s speech or journalist’s article supports their position, one must first figure out if their argument contains any logical flaws independent of one’s world knowledge. Given the prevalence of reasoning independent of world knowledge, we posit that while humans may find statements whose truth values do not align with the real world to be odd, they are not confused by them.
>
> **With regards to MuSR (Sprague et al., 2024), its approach is indeed interesting and relevant to our paper. A brief discussion is included in the Related Works section (L141). We find that its dataset construction method is currently unsuitable for deductive reasoning.**
>
> MuSR’s dataset construction method can be summarized as follows: First, a domain and gold set of facts is manually curated. Second, a reasoning tree is recursively generated using domain-specific prompts on an LLM. Third, a story is generated with GPT-4 and the reasoning tree.
>
> We agree that MuSR achieves prior knowledge independence without contradicting world knowledge because (i) a fictional scenario is manually curated with no connection to the real world, e.g. a murder mystery story, and (ii) GPT-4 is used to generate a complete fictional story.
>
> However, MuSR’s method cannot be applied to deductive reasoning tasks, such as JustLogic. Beyond deductive reasoning, MuSR also tests for inductive reasoning, generalization, and commonsense knowledge (Appendix F, Listing 4 of the MuSR paper). GPT-4 will undeniably make some mistakes during the story generation; the authors of MuSR analyze this in Appendix D. Mistakes include ignoring instructions, hallucination, and invalid logic. Despite the mistakes, the stories are acceptable in MuSR’s task because models can still generalize and inductively reason. However, mistakes will be highly problematic for JustLogic because any imprecision in the premises will make deductive reasoning impossible, therefore rendering the instance useless. If GPT-4 is used to generate JustLogic’s premises, it will no longer be 100% reliable.
>
> Moreover, MuSR’s method is not without downsides. Specifically, MuSR has significantly poorer scalability and diversity. MuSR relies heavily on the human curation of domain, gold facts, and task format, prior to LLM generation. It thus only has 3 domains in the original paper. JustLogic, on the other hand, includes a significantly larger domain due to its use of GenericsKB sentences. With regards to scalability, JustLogic can trivially generate more instances with greater difficulty and domain diversity by running its code. MuSR, on the other hand, requires manual curation to expand into more domains.
>
> (*Further discussion on MuSR is continued in Part 2*)

---

> ### Author Response · Authors · 2024-11-17
> **Response to Reviewer 3iPx (Part 2)**
>
> Nevertheless, thank you for bringing up this paper! MuSR’s approach deserves further exploration as LLMs become more reliable. In our future works, we will study the introduction of LLMs to Step 2 of JustLogic’s dataset construction, while taking deliberate measures to ensure that the validity of each instance is maintained.
>
> **To summarize, given that (i) factually inaccurate statements do not lead to confusion in models and humans, and (ii) alternative methods to ensure prior knowledge independence, e.g. MuSR, are flawed for deductive reasoning, we maintain that our use of factually inaccurate statements is an effective solution to prior knowledge independence.**
>
> > W2: High "natural language complexity", by which the authors mean vocabulary size. I think this is a straightforward overclaim…In my opinion this would also imply moving beyond propositional logic, as natural language largely revolves around predicate-argument structure.
>
> We agree that moving beyond propositional logic and the predicate-argument structure will bring natural language complexity to the next level. **However, while we strive towards this target, this may be too challenging for deductive reasoning benchmarks currently.**
>
> Deductive reasoning benchmarks are typically backed by a formal logic system; the natural language texts are generated or curated based on it. Thus, some level of linguistic rigidity is expected. To illustrate this, below are two sample texts from FOLIO, a human-curated deductive reasoning benchmark, and CLUTRR, a synthetic one. The newly added Appendix D shows more examples and analysis from other benchmarks.
>
> - FOLIO [1]: “All people who regularly drink coffee are dependent on caffeine. People regularly drink coffee, or they don't want to be addicted to caffeine, or both. No one who doesn't want to be addicted to caffeine is unaware that caffeine is a drug…”
> - CLUTRR [2]: “The bald eagle is not rough. The bear does not need the bald eagle. The dog needs the bear. If someone is rough then they chase the bald eagle…”
>
> Evidently, both examples do not reach the natural language complexity of news articles or fiction books. Nonetheless, FOLIO is noticeably more linguistically diverse than CLUTRR. Therefore, in line with the existing literature on deductive reasoning benchmarks [1], we measure natural language complexity based on vocabulary size and the number of domains.
>
> **Given that complexity is evaluated in comparison with other deductive reasoning benchmarks, the claim that JustLogic has high natural language complexity is justified.**
>
> Nonetheless, thank you for drawing attention to this! Improving natural language complexity will be a significant contribution to the deductive reasoning benchmark literature. A discussion regarding this has been added to the Future Works section (Appendix E).
>
> > W3: The 'future-proofness' of the dataset is called into question by the fact that o1-preview outperforms the human average…the authors claim that the dataset can be scaled to match stronger models but do not actually demonstrate this for the strongest model.
>
> We posit the o1-preview’s superior performance to the human average does not call into question the dataset’s ‘future-proofness’.
>
> First, while OpenAI o1-preview outperforms the human average, which includes non-expert participants, it still significantly underperforms the human ceiling (100%). Thus, the model still has substantial margin for improvement.
>
> Second, we demonstrate the JustLogic can be scaled to match stronger models in the error analysis (Section 5.3). In Figure 3 (right), we show that OpenAI o1-preview’s accuracy generally decreases as reasoning depth increases. Llama3-70B and Llama3-8B exhibit similar trends. Thus, to future-proof JustLogic, we can increase reasoning depths to raise the dataset’s difficulty level.
>
> > Suggestion: It would be helpful for clarity if you could use a term other than 'context-independence'...
>
> Thank you for the suggestion! We agree that the term “context” could be confusing because it can be used in two different contexts. The term has been changed to “prior knowledge independence” for better clarity.
>
> > Q1: How is the human ceiling computed?
>
> Human ceiling is measured by the highest accuracy of a single human participant.
>
> [1] Simeng Han, Hailey Schoelkopf, Yilun Zhao, Zhenting Qi, Martin Riddell, Luke Benson, Lucy
> Sun, Ekaterina Zubova, Yujie Qiao, Matthew Burtell, David Peng, Jonathan Fan, Yixin Liu, Brian
> Wong, Malcolm Sailor, Ansong Ni, Linyong Nan, Jungo Kasai, Tao Yu, Rui Zhang, Shafiq Joty,
> Alexander R. Fabbri, Wojciech Kryscinski, Xi Victoria Lin, Caiming Xiong, and Dragomir Radev.
> Folio: Natural language reasoning with first-order logic. arXiv preprint arXiv:2209.00840, 2022.
> URL https://arxiv.org/abs/2209.00840.
>
> [2] Koustuv Sinha, Shagun Sodhani, Jin Dong, Joelle Pineau, and William L Hamilton. Clutrr: A
> diagnostic benchmark for inductive reasoning from text. arXiv preprint arXiv:1908.06177, 2019.

---

> ### Comment · Reviewer_3iPx · 2024-12-02
>
> Thanks for addressing my question and suggestion!
> Your points on prior knowledge independence are well taken. I'm keeping my score the same, as I still think the vocabulary diversity in JustLogic doesn't constitute meaningful linguistic complexity -- I feel existing benchmarks with purely synthetic language i.e. SimpleLogic (Zhang et al., 2022) cover the same ground semantically.

---

> > ### Author Response · Authors · 2024-12-03
> > **Response to Reviewer 3iPx**
> >
> > Thank you for your positive assessment of our response and we appreciate your further comment.
> >
> > > I feel existing benchmarks with purely synthetic language i.e. SimpleLogic (Zhang et al., 2022) cover the same ground semantically.
> >
> > Thank you for raising SimpleLogic in this discussion. The linguistic complexity of synthetic benchmarks, e.g. SimpleLogic, and JustLogic can be straightforwardly compared by examining their sample texts:
> >
> > - SimpleLogic: If messy and hypocritical and lonely, then shiny. If tame, then friendly. If plain and shiny and homely, then nervous. If tender, then hypocritical. If dull and impatient and plain, then tame. If spotless, then perfect. If elegant and tender, then homely…
> > - JustLogic: Either one or both of these statements are true: big head is another sudden death disease which occurs primarily in feedlot cattle, or some energy is transferred by bulbs. The notion that ‘big head is another sudden death disease which occurs primarily in feedlot cattle’ is untrue.
> >
> > A comparison of sample texts from both benchmarks highlights JustLogic’s superior linguistic complexity. JustLogic has (i) more grammatical variations, (ii) more vocabulary diversity, and (iii) better reflects real-world writing.
> >
> > **JustLogic's superior complexity is due to its unique dataset construction method.** Existing synthetic benchmarks rely on templated language and grammar with limited vocabulary. For example, SimpleLogic contains just 150 adjectives. However, JustLogic leverages GenericsKB’s real-world sentences, leading to diverse grammar structures and a large vocabulary size of 10557.
> >
> > JustLogic’s natural language complexity is also demonstrated against other synthetic benchmarks, e.g. CLUTRR, ProofWriter, and ProntoQA, as exhibited in the newly added Appendix D.

---

### Official Review · Reviewer_NSeq · 2024-11-04

**Soundness:** 3
**Presentation:** 3
**Contribution:** 2
**Rating:** 5
**Confidence:** 5

**Summary:**

The paper introduces JustLogic, a benchmark designed to test LLMs' deductive reasoning without the bias of prior knowledge. It highlights the shortcomings of the exitsting logical benchmarks based on complexity and error analysis -- showing some of the sythetic datasets, while may not rely on prior knowledge, are not complex enough. Other datasets, are complex but may be solve using prior human knowledge. Authors curate a large number of logical derivations, using a mix of LLM and human generated templates and propositions from GenericsKB, authors create a logical benchmark, which is synthetic, (possibly) not true in the real world and complex.  Despite improved performance by models like OpenAI o1-preview, they still lag behind human reasoning. JustLogic aims to drive deeper evaluation and enhancement of LLM capabilities.

**Strengths:**

1. The paper highlights flaws in current reasoning benchmarks: low complexity, prior knowledge dependence, and limited error analysis.
2. It addresses the flaws highlighted in point 1 while constructing the new dataset JustLogic that have been unaddressed in datasets like FOLIO, ProofWriter, etc.
3. The solution to the flaws like focusing on argument complexity and Prior knowledge independence looks interesting.
4. Very minimal manual effort is required to construct the dataset.

Authors also aspire to make it more future-proof by leveraging synthetic creation abilities.

**Weaknesses:**

1. There are several points which are claimed but do not have enough justification, for example, it may not be naturally true that a "conclusion" is untrue in the real world just because it is synthetic. No effort have been made to enforce factual inaccuracies -- they should also compare with PrOntoQA, which created a "Fictonal" ontology and a " false" ontology for this purpose. I believe, such a process is required, if you really want to ensure that prior knowledge will not affect the results.
2. Overall the conclusion that every LLM except o1 is lagging -- seems to not add much to the ongoing body of knowledge. Why are some of the derivations difficult vs easy. Only 1/2 line of explanations are given, that too in a hand-wavy manner (L513)
3. There are some issues with authors arguments about "flaws" in the benchmarks. See questions.

**Questions:**

1. Context-ndependent is mentioned in abstract/intro many times. It is unclear how the authors ensured. Seemed like nothing special has been done.
2. Why is the average human performance so low? If you recruit more trained puzzle solvers, such as say linguistic olympiad participants or winners, I believe they may do much better, In that case, this false supremacy of o1 should not be there.
3. L100: Why is it novel? I did not understand what is claimed to be novel here at all.
4. Tab 1: It will be better to clarify what does complexity mean in statistical terms, number of clauses, words etc.
5. L290: This Step 3 is written in a very convouted way or is it the process? Are we saying we have a conclusion and we do not know using logic, whether the conclusion is entailed or not. This can not be true.
Then why a paragraph is randomly assigned an answer? Please explain clearly.
6. L317: How is the number of domains measured? Is it from GenericsKB?
The vocab seems to have been created by a mix of chosen templates and words from GenericsKB. Hard to say that it is significantly complex -- give GKB sentences are quite simple and there are limited templates authors curated.
7. L333: This may not be that simple. One may be able to prove the negation in a smaller set of steps. Is that considered as well? Or, the dataset does not have such examples?
8. L350-353: Very interesting paragraph. Again how do authors ensure the factual incorrectness of the conclusions?
9. L442:Why is the performance on CLUTTRR so low compared to JustLogic?
10. L514: For the apparently long tail argument forms, have you qualitatively evaluated the CoT or some other ways to actually see what reasoning is being done by the LLMs?

---

> ### Author Response · Authors · 2024-11-17
> **Response to Reviewer NSeq (Part 1)**
>
> Thank you for your detailed and constructive reviews! We’re glad that you appreciate how JustLogic addresses the flaws of current reasoning benchmarks.
>
> > W1: …No effort have been made to enforce factual inaccuracies…I believe, such a process is required, if you really want to ensure that prior knowledge will not affect the results.
>
> You are indeed right that some conclusions are factually accurate. However, **it is not true if a conclusion is factually accurate, models may use prior knowledge and therefore lead to artificially high accuracies**.
>
> First, our prompt explicitly instructs models to answer the question only using the paragraph provided, and not to use prior knowledge (Appendix B). Moreover, in few-shot prompts, the examples provided include conclusions where their factual accuracy does not match the correct answer. These measures encourage models to ignore prior knowledge even when conclusions are factually accurate.
>
> Second, we conducted an additional empirical study showing that enforcing factual accuracy is not necessary. The full methodology and results are shown in Appendix C. If factually accurate conclusions cause prior knowledge to affect results, we should find that such questions have higher accuracies.
>
> However, our results show that when reasoning depth is 1, factually inaccurate conclusions in fact exhibit higher performance than factually accurate ones! When reasoning depth is 7 or less, factually inaccurate conclusions exhibit lower performance for Llama3-8B, Llama3-70B, GPT-4, and GPT-4o, but the opposite is true for OpenAI o1-preview. Moreover, the difference in accuracy is typically marginal.
>
> **These results show that the factual accuracy of conclusions has no clear or meaningful impact on accuracy, which suggests prior knowledge independence is upheld regardless. Therefore, enforcing factual inaccuracy is unnecessary to ensure prior knowledge will not affect results.**
>
> **With regards to ProntoQA**: It is indeed an important work in deductive reasoning evaluation. We have included the paper in our related works and the newly added Appendix D, which shows sample texts from various reasoning benchmarks and discusses their linguistic complexity.
>
> Its “fictional” ontology guarantees factual inaccuracy. However, **ProntoQA’s method leads to significantly lower natural language complexity** because each sentence must be programmatically generated rather than extracted from a natural language text database. Moreover, sentences no longer represent real-world scenarios and domains, thus making ProntoQA significantly less realistic than JustLogic.
>
> Here is a sample text from ProntoQA, which illustrates our point: *“Lempuses are bitter. Every lempus is a lorpus. Brimpuses are vumpuses. Tumpuses are impuses. Each impus is not hot. Every numpus is a sterpus. Each shumpus is brown. Sterpuses are fast. Every vumpus is not small…”*
>
> > W2: Overall the conclusion that every LLM except o1 is lagging -- seems to not add much to the ongoing body of knowledge. Why are some of the derivations difficult vs easy…
>
> Thank you for raising this concern! We agree that it is important to extract insights beyond just the lagging performance of LLMs. As such, in our experiment results and error analysis, we show that…
>
> 1. LLMs continue to increase in size in hopes of improving their capabilities, including logical reasoning. However, our results show that increasing model size offers diminishing returns on performance (L459-461). Therefore, we expect that the performance gains in logical reasoning for SOTA models will start to plateau. Alternative methods for improving reasoning should be explored.
>
> 2. Performance improvements offered by increasing model size pale in comparison to those offered by better prompting methods, such as chain-of-thought (L462-465). For example, Llama3-8B with CoT prompting outperforms Llama3-70B with zero-shot prompting. This shows that prompting has an outsized effect on reasoning capabilities and that the specific mechanisms that cause this deserve further study.
>
> 3. OpenAI o1-preview outperforms the second-best model, GPT-4o, by 15 percentage points, which is a significant improvement. This proves that the former’s approach of using reinforcement learning and CoT prompting leads to substantially better deductive reasoning in LLMs (L466-468). To the best of our knowledge, this paper is the first to evaluate and highlight OpenAI o1-preview’s deductive reasoning abilities.
>
> *(Points 4 and 5 are found in Part 2)*

---

> > ### Author Response · Authors · 2024-11-17
> > **Response to Reviewer NSeq (Part 2)**
> >
> > 4. The accuracies of some argument forms are significantly better than others (L513-518). This is explained by the fact that some argument forms are more commonly used than others. For example, modus ponens (‘If x, then y. x. Therefore, y’) is extremely common, while absurdum ad reductio is rarely used. This explains why the former exhibits higher accuracy than the latter.
> >
> > 5. Model accuracies generally decrease as reasoning depth increases (L519-L527). The phenomenon is quite intuitive: arguments with higher depth are more complex and therefore harder to solve correctly. However, empirically demonstrating this is crucial to validate JustLogic’s ‘future-proofness’ (Section 3.5). As LLMs become stronger, JustLogic can be adapted to match them by increasing reasoning depth, which we have shown to result in lower accuracies.
> >
> > > Q1: Context-ndependent is mentioned in abstract/intro many times. It is unclear how the authors ensured. Seemed like nothing special has been done.
> >
> > There seems to be confusion about what context-independence is. This is defined in L56, which states “we developed a novel test for context-independence, which measures the influence of prior knowledge on reasoning benchmarks.” In other words, the “prior knowledge independence” you mention in Strengths 1 and 3 is exactly the same as context-independence.
> >
> > However, we acknowledge that the term “context-independence” may be confusing and that “prior knowledge independence” is a better term. The paper has therefore been revised to use the term “prior knowledge independence” instead.
> >
> > > Q2: Why is the average human performance so low? If you recruit more trained puzzle solvers, such as say linguistic olympiad participants or winners, I believe they may do much better, In that case, this false supremacy of o1 should not be there.
> >
> > You are right that more trained human annotators will perform better on JustLogic. In fact, our human evaluation does include expert participants, which is reflected in the human ceiling performance of 100%.
> >
> > At the same time, the inclusion of non-expert participants is also valuable. By including participants with varying levels of expertise, we can more comprehensively examine human performance on JustLogic. For example, non-experts are less familiar with how to make valid arguments, so they are more likely to arrive at the wrong conclusions. This is reflected in the lower average human accuracy.
> >
> > > Q3: L100: Why is it novel? I did not understand what is claimed to be novel here at all.
> >
> > Prior knowledge independence ensures model performance reflects reasoning capabilities, rather than comprehensive world knowledge. However, its importance is overlooked by existing deductive reasoning benchmarks; they do not robustly prove that their datasets are independent of prior knowledge.
> >
> > We therefore formulate this test which empirically measures whether a dataset is independent of prior knowledge (Section 4.1). This test has never been conducted before and will help facilitate prior knowledge independence in future benchmarks.
> >
> > > Q4: Tab 1: It will be better to clarify what does complexity mean in statistical terms, number of clauses, words etc.
> >
> > Natural language complexity in statistical terms is defined in L317-319. The statistics of each comparison benchmark are reflected in Table 4.
> >
> > > L290: This Step 3 is written in a very convoluted way or is it the process?
> >
> > Thank you for the suggestion! We have improved the explanation of Step 3 in Section 3.3. The new paragraph is as follows:
> >
> > *“The LLM's task is to determine whether the given query statement is true, false, or uncertain based on the premises provided. Using Figure 2 as an example, if we assign the query statement to be the negation of the conclusion, i.e. "It is not true that Japan is in Asia", then the answer is false. If the query statement is the same as the conclusion, then the answer is true. If the query statement is unrelated to the premises, then the answer is uncertain.”*
> >
> > > L317: How is the number of domains measured? Is it from GenericsKB? The vocab seems to have been created by a mix of chosen templates and words from GenericsKB. Hard to say that it is significantly complex -- give GKB sentences are quite simple and there are limited templates authors curated.
> >
> > Regarding how the number of domains is measured: Yes, the domain of each sentence is provided by GenericsKB.
> >
> > Regarding the natural language complexity of JustLogic: (*Continued in Part 3*)

---

> ### Author Response · Authors · 2024-11-17
> **Response to Reviewer NSeq (Part 3)**
>
> Regarding the natural language complexity of JustLogic: We hold that GenericsKB sentences and our templates are sufficiently complex. GenericsKB appears simple but actually contains highly diverse vocabulary and sentence structures. It extracts 1.7B sentences from Waterloo, SimpleWiki, and ARC, then selects 3.4M generic, standalone statements using lexicosyntactic rules and a BERT classifier. Moreover, many other datasets utilize sentences from GenericsKB [1,2,3], which suggests it has sufficient linguistic complexity to be used in datasets.
>
> As for our templates, we believe you are referring to the expressions of various logical forms (Table 3), of which there are 50 in total. This introduces significant diversity because many statements in JustLogic are compositions of multiple logical forms. For example, the logical form x → ¬y contains a conditional and negation, which has 11 and 15 unique expressions respectively. Thus, this logical form has 11x15=165 possible expressions. Moreover, x and y are randomly chosen from GenericsKB’s 3.4M sentences.
>
> Therefore, the JustLogic dataset has high natural language complexity.
>
> > L333: This may not be that simple. One may be able to prove the negation in a smaller set of steps. Is that considered as well? Or, the dataset does not have such examples?
>
> The solutions provided by JustLogic are already the fastest way to arrive at the conclusion. This is because of how the argument structure is constructed (L252-259): the LLM must reproduce exactly the same argument forms in the same order to derive the correct solution. Therefore, the reasoning depth of each instance is well-defined.
>
> > L350-353: Very interesting paragraph. Again how do authors ensure the factual incorrectness of the conclusions?
>
> Please refer to our response to W1, where we show that the factual correctness of the conclusions does not affect prior knowledge’s influence on the results.
>
> > L442:Why is the performance on CLUTTRR so low compared to JustLogic?
>
> That is because CLUTRR has 16 answer options while JustLogic only has 3, therefore making it harder for the model to answer correctly. However, JustLogic’s performance is in fact better because its accuracy is lower than random probability which indicates that prior knowledge is not useful for answering the question. CLUTRR, on the other hand, has a higher accuracy than random.
>
> > L514: For the apparently long tail argument forms, have you qualitatively evaluated the CoT or some other ways to actually see what reasoning is being done by the LLMs?
>
> We have considered qualitatively evaluating the LLMs’ reasoning. However, this requires considerable effort because different models structure their CoT responses differently. In fact, different questions within the same LLM can result in vastly different forms of CoT. Thus, every response must be manually and individually parsed to identify where exactly the model failed.
>
> Moreover, based on our examination of several LLM responses, there is a wide and complex array of potential errors, including hallucinated premises, not using relevant premises, incorrect usage of argument forms, and even responses that are simply incoherent. Such an analysis requires careful and extensive study, which is beyond the scope of this paper.
>
> The error analysis (Section 5.3), where we study how reasoning depth and argument form affect accuracy for various models, is the first step toward a more thorough investigation of how exactly LLMs reason.
>
> Nevertheless, this is an interesting and promising direction, which we intend to explore in the future. A discussion on this has been added to the Future Works section (Appendix E). Thank you for the suggestion!
>
>
> [1] Bill Yuchen Lin, Seyeon Lee, Rahul Khanna, and Xiang Ren. 2020. Birds have four legs?! NumerSense: Probing Numerical Commonsense Knowledge of Pre-Trained Language Models. In Proceedings of the 2020 Conference on Empirical Methods in Natural Language Processing (EMNLP), pages 6862–6868, Online. Association for Computational Linguistics.
>
> [2] Hao Peng, Xiaozhi Wang, Shengding Hu, Hailong Jin, Lei Hou, Juanzi Li, Zhiyuan Liu, and Qun Liu. 2022. COPEN: Probing Conceptual Knowledge in Pre-trained Language Models. In Proceedings of the 2022 Conference on Empirical Methods in Natural Language Processing, pages 5015–5035, Abu Dhabi, United Arab Emirates. Association for Computational Linguistics.
>
> [3] Li Du, Xiao Ding, Kai Xiong, Ting Liu, and Bing Qin. 2022. e-CARE: a New Dataset for Exploring Explainable Causal Reasoning. In Proceedings of the 60th Annual Meeting of the Association for Computational Linguistics (Volume 1: Long Papers), pages 432–446, Dublin, Ireland. Association for Computational Linguistics.

---

### Author Response · Authors · 2024-11-17
**General Response to Reviewers**

## Summary

Thank you to all reviewers for your thoughtful reviews and feedback! We appreciate your recognition of the significant shortcomings in existing deductive reasoning benchmarks and your positive assessment of how JustLogic solves them through its novel dataset construction method.

We respond to individual reviews below. This general response summarizes the main changes to the paper. All changes are colored blue.

## Highlighting JustLogic’s contributions

We further clarified JustLogic’s contributions to the literature by refining the introduction (L66-71) and conclusion (L530-538).

Logical reasoning is one of the key challenges for LLMs. Despite great advances made in recent years, it remains controversial whether deductive reasoning abilities have genuinely progressed. This is due to the lack of a reliable benchmark that accurately and comprehensively evaluates deductive reasoning. JustLogic timely addresses this critical research gap.

**Importance of a reliable deductive reasoning benchmark**: Existing benchmarks used to evaluate LLMs’ reasoning abilities obfuscate their true deductive reasoning capabilities. This is because they simultaneously evaluate many types of reasoning, e.g. inductive, deductive, and analogical, and involve many types of knowledge, e.g. commonsense, math, and scientific knowledge (Section 2.1).

In response, benchmarks that solely test for deductive reasoning have attracted considerable interest. However, their reliability is questionable due to several fundamental issues, i.e. prior knowledge dependence, lack of complexity, and lack of error analysis (Table 1).

**JustLogic** addresses the limitations of existing benchmarks. Specifically, we ensure…
- Prior knowledge independence by generating arguments that do not align with real-world knowledge. We prove this via our prior knowledge independence test.
- Argument complexity by synthetically generating argument structures
- Natural language complexity by leveraging GenericsKB sentences and curated templates
- Error analysis capabilities due to JustLogic’s high degree of flexibility, e.g. ability to adjust reasoning depth.

## Change of Terminology

The term “context-independent” has been changed to “prior knowledge independence” for better clarity. The term "context" now exclusively refers to the paragraph of premises provided by each JustLogic question.

## New Experiment: Impact of Factual Accuracy on Model Performance (Appendix C)

We conducted a new empirical study to investigate whether the factual accuracy of conclusions affects model performance. This is highly relevant to comments by reviewers NSeq, 3iPx, and LXqP.

The results show that there is no consistent relationship between factual accuracy of conclusions and model performance. The evidence does not support the idea that inaccurate conclusions confuse models and factually accurate conclusions give models an advantage.

This is because (i) our prompts explicitly ask models to avoid prior knowledge, and (ii) how LLMs treat factual accuracy when reasoning deductively depends on the LLM’s training. Further, we posit that the ability to perform deductive reasoning while ignoring prior knowledge is a fundamental human skill, and models should be evaluated in this regard.

Therefore, we conclude that the inclusion of both factually accurate and inaccurate conclusions is a justified way of ensuring prior knowledge independence. Further details on the experiment and subsequent analysis can be found in Appendix C.

## New Analysis: Sample Texts from Deductive Reasoning Benchmarks (Appendix D)

We recognize that the metrics for natural language complexity, i.e. vocabulary size and number of domains, do not fully capture the range of linguistic patterns in datasets. Thus, we show sample texts from CLUTRR, ProofWriter, ProntoQA-OOD, LogiQA 2.0, FOLIO, and JustLogic. This is relevant to reviewers NSeq, 3iPx, and LXqP.

By examining the texts, we highlight JustLogic’s superior or comparable natural language complexity to other reasoning benchmarks.

## New Section: Future Works (Appendix E)

The reviewers provide insightful comments on how JustLogic can be further developed. While these are beyond the scope of this work, we intend to explore these ideas further in the future. We therefore outline the following in the Future Works section:
1. Further enhance JustLogic’s natural language using LLM-generated text, as suggested by reviewer 3iPx.
2. More comprehensive error analysis on how and where exactly LLMs fail, as suggested by reviewers NSeq and LXqP.
3. Scaling JustLogic to incorporate more logical reasoning-related question types.

---

### Comment · Area_Chair_cBf3 · 2024-11-25
**Action Required: Respond to Author Rebuttals - Nov 27**

Dear ICLR Reviewers,

The author discussion phase is ending soon. Please promptly review and respond to author rebuttals for your assigned papers. Your engagement is critical for the decision-making process.

Deadlines:

November 26: Last day for reviewers to ask questions to authors.
November 27: Last day for authors to respond to reviewers.
November 28 - December 10: Reviewer and area chair discussion phase.
Thank you for your timely attention to this matter.

---

### Meta-Review · Area_Chair_cBf3 · 2024-12-21

**Metareview:**

The paper introduces JustLogic, a synthetically generated benchmark designed to test LLMs' deductive reasoning capabilities while eliminating the influence of prior knowledge. The authors evaluate various LLMs including OpenAI o1-preview on JustLogic, achieving 81% accuracy compared to a human average of 73%.

The reviewers value the paper's motivation to address the flaws in current reasoning benchmarks and its comprehensive evaluation with multiple LLM models. Key concerns discussed include: the decision to use factually inaccurate statements (though authors showed empirically this does not affect model performance), limited investigation of failure modes in the error analysis, and questions about whether vocabulary diversity truly constitutes meaningful linguistic complexity compared to existing synthetic benchmarks such as SimpleLogic.

The authors have engaged constructively with reviewer feedback, providing additional empirical analysis on factual accuracy effects and clarifying their benchmark design choices. While some concerns about error analysis depth and linguistic complexity remain to be addressed in future work, I give a rejection as the paper's contribution to deductive reasoning evaluation, while valuable, needs further development in terms of thoughtful error analysis and linguistic complexity before meeting ICLR's standards.

**Additional Comments On Reviewer Discussion:**

Already included in the meta review.

---

### Decision · Program_Chairs · 2025-01-22

Reject